# Feasibility and Efficacy of Low-to-Moderate Intensity Aerobic Exercise Training in Reducing Resting Blood Pressure in Sedentary Older Saudis with Hypertension Living in Social Home Care: A Pilot Randomized Controlled Trial

**DOI:** 10.3390/medicina59061171

**Published:** 2023-06-18

**Authors:** Abdulrahman A. Alzahrani, Abdulfattah S. Alqahtani, Vishal Vennu, Saad M. Bindawas

**Affiliations:** 1Social Care Center for the Elderly, Mecca 12840, Saudi Arabia; 2Department of Rehabilitation Sciences, College of Applied Medical Sciences, King Saud University, Riyadh 11433, Saudi Arabia; 3King Salman Center for Disability Research, Riyadh 11614, Saudi Arabia

**Keywords:** hypertension, aerobic exercise, blood pressure, heart rate, body fat, cholesterol, social home care

## Abstract

*Background and Objectives*: The effect of non-pharmacological aerobic exercise training on blood pressure in sedentary older individuals receiving social home care in Saudi Arabia has not been investigated. This study aimed to examine the effects of aerobic exercise on blood pressure in sedentary older Saudis with hypertension residing in these settings. *Materials and Methods*: A pilot randomized control trial was conducted with 27 sedentary individuals, aged 60–85, diagnosed with hypertension, and living in social home care in Makkah, Saudi Arabia. Recruitment took place between November 2020 and January 2021, and participants were randomly assigned to either the experimental or control group. The experimental group engaged in three 45 min sessions of low-to-moderate intensity aerobic activity per week for eight weeks. This trail was registered with the ISRCTN registry (ISRCTN50726324). *Results*: Following eight weeks of mild to moderate aerobic exercise training, the primary outcome of resting blood pressure showed a significant reduction in the experimental group (systolic blood pressure: mean difference [MD] = 2.91 mmHg, 95% confidence interval [CI] = 1.61, 4.21, *p* = 0.001; and diastolic blood pressure: MD = 1.33 mmHg, 95% CI = 1.16, 1.50, *p* = 0.001) compared to the control group. Within the experimental group, there was also a significant decrease in systolic blood pressure (MD = −2.75 mmHg, 95% CI = −7.73, 2.22, *p* = 0.005) and diastolic blood pressure (MD = −0.83 mmHg, 95% CI = −5.81, 4.14, *p* = 0.02). *Conclusions*: This trial demonstrates the feasibility and potential benefits of low-to-moderate intensity aerobic exercise training in reducing resting blood pressure among sedentary older Saudis with hypertension residing in this aged care setting.

## 1. Introduction

Hypertension is strongly associated with age [1]. Approximately 70% of older adults live with high blood pressure, especially in nursing homes [2]. About 80% of older adults who reside in nursing homes spend their time being inactive or in a seated position [3,4]. This sedentary behavior in nursing home residents significantly leads to the development of obesity and hypertension, which contribute to the risk of cardiovascular diseases and cause premature death [5].

In Saudi Arabia, hypertension affects 26% of the adult population and is a major cause of death [6]. Approximately 24% of all deaths are due to cardiovascular conditions [6]. Hence, within the realms of health care, particularly in nursing home environments, the prioritization of optimal blood pressure management emerges as a crucial non-pharmacological approach [7,8]. Existing evidence on the effect of exercise on blood pressure in patients with resistant hypertension shows that intervention programs with 3 sessions per week for 8–12 weeks significantly reduces 24 h systolic blood pressure by 9.9 mmHg and diastolic blood pressure by 5 mmHg [9]. It has been reported that a reduction in systolic and diastolic BP of 2 mmHg reduces cerebrovascular accident risk by 14% to 17% and ischemic heart disease risk by 6% to 9% [10].

Given the considerable risk associated with hypertension, the American College of Sports Medicine recommends regular aerobic exercise as a non-pharmacological treatment strategy for hypertension [11]. Several clinical trials and meta-analyses have demonstrated that regular aerobic exercise prevented the development of hypertension by reducing and controlling an individual’s blood pressure [7,12]. For example, a previous study investigated the effect of aerobic exercise training on blood pressure using 72 randomized control trials involving 3936 adult participants [13]. This study found a net reduction of 6.9 mmHg in systolic blood pressure and 4.9 mmHg in diastolic blood pressure.

Clinical trials investigating the effect of aerobic exercise in older adults have reported divergent results [14]. To our knowledge, no previous study has examined the impact of aerobic exercise on the BP of sedentary older adults living in social home care in Saudi Arabia [15]. This pilot study aimed to investigate the feasibility and efficacy of non-pharmacological aerobic exercise training on resting blood pressure, heart rate, body fat, cholesterol, and functional status in sedentary older adults with hypertension residing in social home care setting. The hypothesis was that implementing a non-pharmacological aerobic exercise program would be feasible and result in positive changes in resting blood pressure and other outcomes among sedentary older adults in this aged care setting.

## 2. Materials and Methods

We conducted this pilot study using a nonblinded randomized controlled trial (RCT) design at the Medical Department of Social Home Care for older adults in Makkah, Saudi Arabia, from January 2021 to March 2021. The study was conducted in accordance with the Declaration of Helsinki and approved by the Institutional Review Board of King Saud University (E-20-5451) and the Ministry of Human Resource and Social Development, Social Welfare and Family Agency (78891). This study was registered with the ISRCTN registry (ISRCTN50726324). We opted for a nonblinding RCT design based on the recommendation that blinding should not be a decisive factor in assessing bias in a rehabilitation context [16]. We reviewed the medical records of 65 individuals living in social home care to identify those diagnosed with hypertension and determine potential participants for the study. The examiner completed a screening form to decide whether they could participate in the study. The examiner described the investigation process to the participant who fits the following study inclusion and exclusion criteria.

Sedentary men or women, aged 60 to 85, with hypertension and living in social home care were eligible. We defined hypertension as systolic blood pressure ≥130 mmHg and diastolic blood pressure ≥80 mmHg [17]. We described a sedentary lifestyle as having no engagement with sports or regular exercise during the previous three months [18]. The Saltin–Grimby Physical Activity Level Scale (SGPALS) assessed the participants’ sedentary behavior. The SGPALS is a valid and reliable instrument for objectively assessing individual physical activity levels [19]. We evaluated the ability to do the dynamic exercise using the Borg rating of preserved exertion scale [20]. We excluded participants if they had heart failure, a history of unstable angina or myocardial infarction, chronic pulmonary disease, or major musculoskeletal disorders.

We randomly divided participants who complied with the inclusion criteria into experimental and control groups, as shown in Appendix A. We used a computer software that created a random sequence to allocate the two groups. We collected all data at the start of the intervention and after the eight-week period. We collected data 48 h after the last exercise session to avoid the post-exercise hypotension effect. Usually, post-intervention measurements were taken at the same time of day and following the same procedure used in the baseline. Reliability assessments were carried out for the examiner nurse, who measured blood pressure and heart rate intra-rater reliability to avoid variation in blood pressure value.

We assessed the primary and secondary outcomes at baseline and after the eight-week aerobic exercise training program. We determined the primary outcome of resting blood pressure by evaluating systolic and diastolic blood pressure using a conventional mercury sphygmomanometer (Rossmax Swiss GmbH, Heerbrugg, Switzerland). To assess systolic and diastolic blood pressure, we instructed all participants to cease smoking and drinking caffeine beverages 10 h before measurement and to empty their bladder. Participants were allowed to sit comfortably on a chair for at least 15 min with the back straight, supported, feet flat on the floor, legs not crossed, and arms supported on a flat surface. Blood pressure was measured twice, two minutes apart, and on two different occasions from both arms. We used the highest reading from the mean of the two readings as the blood pressure value. The mercury sphygmomanometer is valid, reliable, and considered the gold standard for measuring blood pressure [17].

The secondary outcomes were heart rate, body fat, total cholesterol, low-density lipoprotein (LDL), high-density lipoprotein (HDL), and functional status. We assessed the resting heart rate using heart rate monitors when the participants were calm, relaxed, sitting, or lying. The term “resting heart rate” describes the heartbeat when it pumps the least blood required at rest. Heart rate monitors detect and measure heart or pulse rate and have proven to be accurate and valid tools [21]. Body fat was determined using body mass index (BMI). The BMI was determined manually using the metric BMI formula [22] as weight in kilograms divided by height in meters squared. We measured the participants’ weight without shoes or heavy outer clothes using a standard stadiometer, and their height was measured in the standing position. We evaluated the participants’ total cholesterol, LDL, and HDL levels by analyzing the blood sample in the laboratory. The functional ability was measured using the Katz index scale [23].

According to the American College of Sports Medicine guidelines [24], we instructed participants in the experimental group to perform mild to moderate intensity aerobic exercise involving 24 sessions lasting 45 min 3 times per week for eight weeks using stationary cycling under the physiotherapist’s supervision. Using stationary bicycles is an efficient and effective method of burning calories and body fat while enhancing the heart, lungs, and muscles compared to other cardio equipment. While putting less strain on your joints, a stationary bike provides one with benefits of great aerobic exercises [25]. Participants must attend more than 90% of all training sessions at the physiotherapy gym in the Medical Department of Social Home Care. The eight-week program in this study was based on previous descriptions [26,27] and on the recommendation that every adult accumulates at least 30 min of moderate physical activity on most days of the week [28]. Furthermore, a previous epidemiologic study found that people who exercised at least an hour per week had about half the risk of coronary heart disease as opposed to those who did not [29].

In the first week, participants started from 20 min with 30% of their heart rate and progressed gradually to 45 min with 50%. For the duration of each session, a 10 min warm-up and cool-down routine was included. We used a heart rate monitor to maintain aerobic exercise intensity during training and prevent excessively elevated heart rate of a maximum of 160 beats per minute (bpm) [30] under the physiotherapist’s supervision. Participants in the control group received the usual care. For ethical reasons, participants in the control group were invited to join the aerobic exercise program after completing the eight-week intervention period. The adherence of participants to the exercise sessions was reported. Both groups used antihypertensive medications.

The study sample size was calculated based on the inconsistent results of RCTs conducted on older adults in different settings and the divergent conclusions of the meta-analyses findings [31,32], a reduction of 3 ± 1.2 mmHg in systolic pressure and 1 ± 0.81 mmHg in diastolic pressure. A significant decrease was detected using an alpha of 0.05, a power of 0.8, and an effect size of 0.5. The required total sample size was 40 participants for this pilot study.

We evaluated the dependent variables’ normality distribution and homogeneity of variance using the Kolmogorov–Smirnov and Levene’s tests, respectively. Descriptive statistics were presented as mean and standard deviation (SD) in total and for both experimental and control groups. The change in both the primary and secondary outcomes between the groups was analyzed using a nonparametric statistical test of the Mann–Whitney U test because of the limited sample size of this study. The change in the primary and secondary outcomes within groups was analyzed using a nonparametric statistical test of the Wilcoxon signed-rank test. Significance between groups was determined using an independent t-test, whereas a dependent sample test was used for within-group significance determination. The statistics were presented in mean and standard deviations (SD), along with 95% confidence interval (95% CI). All analyses were performed using IBM SPSS software (IBM, SPSS Inc., Armonk, NY, USA). Statistical significance was defined as *p*-value < 0.05.

## 3. Results

We excluded 35 patients out of the 65 who were not diagnosed with hypertension (systolic and diastolic blood pressures of 130 and 80 mmHg, respectively) [17]. Patients with a below-knee amputation (*n* = 1), older than 85 years old (*n* = 1), and with chronic obstructive pulmonary disease (*n* = 1) were also excluded. Finally, the study included a total of 27 participants. Due to 3 dropouts, only 24 of the 27 participants initially enrolled in the study—12 in each experimental and control group—were used in the final analysis. The reasons for dropping out were relocation, follow-up loss, and trial withdrawal (Figure 1).

Participants in the experimental group were approximately one year older, overweight (26.1 vs. 24.6), and had a higher average LDL compared to the control group (103.2 vs. 79.3). Both groups had an average heart rate of 81 bpm (Table 1).

After eight weeks of mild to moderate aerobic exercise training, resting blood pressure was significantly reduced between groups (systolic blood pressure: mean difference [MD] = 2.91 mmHg, *p* = 0.001; and diastolic blood pressure: MD = 1.33 mmHg, *p* = 0.001). Within-group resting blood pressure was also significantly reduced (systolic blood pressure: MD = −2.75 mmHg, *p* = 0.005; and diastolic blood pressure: MD = −0.83 mmHg, *p* = 0.02). No significant changes occurred in the mean of systolic and diastolic blood pressures among the control group (*p* > 0.05) (Table 2).

The mean heart rate, body fat, cholesterol, and LDL cholesterol were significantly decreased between groups (MD = 3.83 beats/min (*p* = 0.002), 0.81 kg/m^2^ (*p* = 0.001), 40.9 mg/dL (*p* = 0.001), and 50.3 mg/dL (*p* = 0.001), respectively). No significant changes occurred between groups in the mean HDL and functional status (*p* = 0.078 and *p* = 1.00, respectively). The mean heart rate, body fat, cholesterol, and low-density lipoprotein (LDL) cholesterol were significantly decreased within-group (MD = −3.67 beats/min (*p* = 0.005), −0.71 kg/m^2^ (*p* = 0.009), −34.0 mg/dL (*p* = 0.002), and−46.0 mg/dL (*p* = 0.002), respectively). No significant change occurred in the mean of HDL and functional status in within-group (*p* = 0.929 and *p* = 1.00, respectively). No significant changes occurred in the mean of resting heart rate, BMI, cholesterol, LDL, HDL, and functional status from pre-to-post exercise among the control group (*p* > 0.05) (Table 3).

## 4. Discussion

Results indicated that resting blood pressure was significantly reduced between and within-group after eight weeks of mild to moderate aerobic exercise training. No significant changes occurred in the mean of resting systolic and diastolic blood pressure, resting heart rate, BMI, cholesterol, LDL, HDL, and functional status from pre-to-post exercise among the control group.

In this study, the systolic and diastolic blood pressure reduction following aerobic exercise was less than expected (−2.75 mmHg and −0.83 mmHg, respectively). However, decrease in the systolic and diastolic blood pressures might be the underlying physiological process that occurs due to the increasing shear stress in the blood vessels, which increases the formation of nitric oxide (NOx) vasodilator and contributes to vascular vasodilatation [33]. An earlier trial with 71-year-old volunteers who were overweight, sedentary, hypertensive, and hypercholesterolemic discovered similar outcomes with a higher reduction in systolic blood pressure (−10 ± 5 mmHg) and diastolic blood pressure (−5 ± 2 mmHg) [34]. A possible reason for the greater reduction in blood pressure may be because their intervention was three months of high-intensity vigorous aerobic exercise. Another previous study also reported a greater reduction in systolic and diastolic blood pressures by −7.7 and −5 mmHg, respectively, in the exercise group with a mean age of 52 years [35]. These results might be because of the three months longer interventions conducted with more frequency (five times/week). Although all NOx interact during exercise, as shown by several studies [36,37], there is not enough data to prove or disprove the use of inorganic nitrate for lowering blood pressure [38]. More study on the relationship between all NOx and blood pressure is necessary due to the gaps in the literature and the above studies’ limitations.

The findings of this study agree with those of a recent meta-analysis and systematic review of clinical trial studies [15]. According to this study, regular exercise significantly lowers systolic and diastolic blood pressure by 4.0 mmHg in older people with hypertension. Another recent study that showed mild to moderate intensity aerobic exercise using a treadmill in diabetic older adults without functional impairments decreased the systolic and diastolic blood pressure by –6.3 mmHg and −3 mmHg, respectively [39]. However, the reasons for less than a predicted reduction in resting systolic and diastolic blood pressures in this study can be attributed to critical methodological differences from the previous three studies mentioned above, including different participant characteristics. The participants of the earlier studies were active and mobilized older adults living in the community without functional impairments.

The secondary outcomes of this study showed that aerobic exercise effectively reduces resting heart rate, BMI, cholesterol, and LDL. These findings are broadly consistent with previous studies. For example, Madden et al. (2009) [34] reported a reduction in resting heart rate for the experimental group by −5 beats/min. A possible reason for the decrease in resting heart rate might be the beneficial physiological effect of regular aerobic training that causes a decline of the norepinephrine plasma level in the bloodstream, along with sympathetic activation reduction [40]. Stewart et al. (2005) [41] found that regular aerobic exercise decreases BMI and weight by −0.8 kg/m^2^ and 2.3 kg, respectively. Blumenthal et al. (2000) [42] reported that body weight reduction is associated with decreased blood pressure. In addition, Higashi et al. (1999) [35] reported a significant decrease in cholesterol and LDL in the exercise group.

In this study, the results showed that aerobic training does not improve functional ability. This finding is similar to a previous study that reported no improvement in functional status after 12 weeks of aerobic exercise [43]. This result may have occurred because of the mild to moderate aerobic exercise intensity intervention and a focus on the lower extremity.

This study has several strengths. To our knowledge, this is the first study in Saudi Arabia investigating the beneficial effects of regular aerobic exercise on resting blood pressure in sedentary older adults with hypertension living in social home care. Nurse assessors who measured study participants’ blood pressure were blinded for allocation to avoid measurement bias. The physiotherapist and attendance supervised the exercise sessions, and the experimental group participants’ adherence was accomplished excellently. Additionally, all exercise sessions had the correct training time and amount of intensity being completed using heart rate monitors.

However, this study has several limitations that should be acknowledged as well. Firstly, it is worth noting that the trial was conducted with a relatively small sample size of 27 sedentary older Saudis with hypertension residing in a social home care. While the study yielded significant findings, the limited number of participants may restrict the generalizability of the results to a larger population. Therefore, future studies with larger sample sizes are necessary to confirm and strengthen the observed effects. Additionally, it is crucial to account for attrition rates and consider the potential influence of the COVID-19 pandemic on participant dropout during the study period. Secondly, the study was exclusively conducted among sedentary older Saudis with hypertension residing in a specific aged-care setting in Makkah, Saudi Arabia. While this design allowed for a focused investigation, it may limit the generalizability of the findings to other populations, including individuals from different regions, cultural backgrounds, or socio-economic status. To ensure the external validity of the results, future research should aim to include a more diverse participant pool. Another limitation is the relatively short duration of the aerobic exercise training intervention, which lasted for eight weeks. While this duration was sufficient to demonstrate significant reductions in resting blood pressure, it is possible that longer intervention periods could yield different or more pronounced effects on blood pressure control. Future studies should explore the impact of extended exercise training durations to evaluate the sustained effects on blood pressure in sedentary older individuals. Moreover, it should be noted that this study focused exclusively on cycling as a form of aerobic exercise intervention. Although cycling is a popular and effective aerobic activity, the findings may not directly apply to other aerobic exercises or physical activity types. Incorporating interventions involving different forms of aerobic exercises, such as walking or swimming, would enhance the comprehensiveness and generalizability of the study. Future research should consider evaluating the effects of various aerobic exercise modalities on blood pressure in sedentary older individuals with hypertension. Additionally, another limitation of this trial is the need for long-term follow-up assessments. This study focused solely on the immediate effects of aerobic exercise training on resting blood pressure. The lack of long-term follow-up hinders our understanding of the durability of the observed reductions in blood pressure over an extended period. Subsequent studies should include long-term follow-up periods to evaluate the maintenance of blood pressure improvements and assess the sustainability of the intervention’s effects. Furthermore, despite efforts to minimize confounding factors, it is vital to acknowledge the possibility of other unmeasured variables influencing the observed outcomes. Factors, such as dietary changes, medication adjustments, and other lifestyle modifications were not explicitly controlled for, which may have influenced the results. Future studies should incorporate comprehensive assessments, including detailed dietary and medication records, to better understand and control potential confounders. Lastly, while the study demonstrated the feasibility and efficacy of aerobic exercise training, it is important to acknowledge that cycling may only be suitable or preferred by some sedentary older individuals. Factors, such as physical limitations, personal preferences, and accessibility to cycling facilities may impact the applicability of the findings to a broader population. Future research should explore and evaluate the feasibility and efficacy of different types of aerobic exercises to provide a range of options for sedentary older individuals with hypertension. Acknowledging these limitations is crucial for comprehensively understanding the study’s scope and potential implications. Future research endeavors should address these limitations to enhance the findings’ validity, generalizability, and clinical applicability in managing hypertension among sedentary older individuals residing in social home care settings.

The findings of this study may have broad clinical implications because the results reflected a significant reduction in resting systolic and diastolic blood pressure, heart rate, BMI, and cholesterol in the exercise group. Physiologically, reducing cholesterol and LDL concentration in blood circulation may decrease vascular endothelial cell damage, thereby decreasing atherosclerosis and vascular stiffness, which are significant causes of hypertension and coronary artery disease [44], particularly in older adults [45]. Thus, it can be concluded that regular aerobic exercise is a viable and efficient non-pharmacological treatment plan to employ successfully in older adults with hypertension [15], especially in the post-COVID era in nursing home settings. Developing targeted exercise interventions for sedentary older adults can play a crucial role in reducing hypertension and improving cardiovascular health in this vulnerable population. By promoting physical activity and exercise as a non-pharmacological approach, healthcare providers can contribute to the overall well-being and quality of life of sedentary older adults with hypertension.

## 5. Conclusions

In conclusion, our pilot trial provides evidence supporting the feasibility and efficacy of regular aerobic exercise as a non-pharmacological approach to reduce resting blood pressure and associated risk factors in sedentary older adults with hypertension in Saudi Arabia. It is essential to conduct further research on a national scale to enhance the generalizability of these findings. Future investigations should incorporate a more comprehensive statistical analysis, larger sample sizes, and explore alternative exercise modalities tailored specifically to this population. Additionally, long-term studies are necessary to establish the sustained impact of regular aerobic exercise on resting blood pressure. These findings significantly impact the development of targeted exercise interventions to reduce hypertension and improve cardiovascular health among sedentary older adults, particularly in nursing homes and in the post-COVID era.

## Figures and Tables

**Figure 1 medicina-59-01171-f001:**
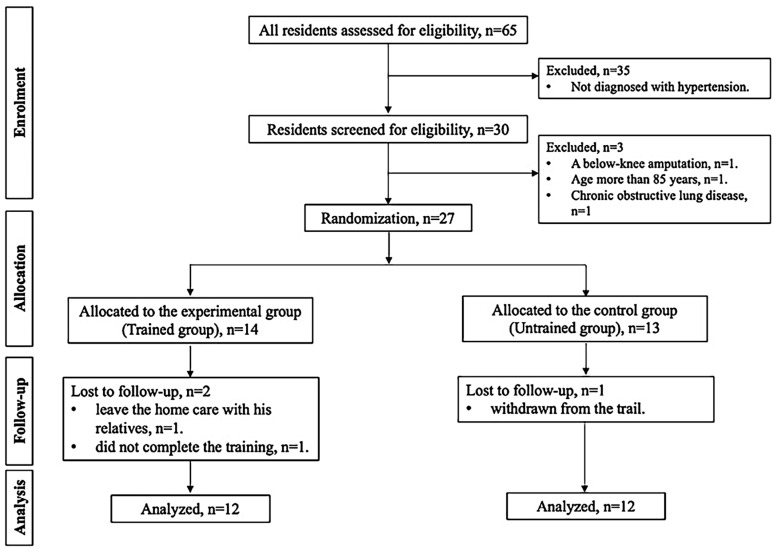
The flowchart of the study participants.

**Table 1 medicina-59-01171-t001:** Descriptive characteristics of study participants at baseline.

	All Participants	Experimental Group	Control Group
N	24	12	12
Sex, F/M, *n*	5/19	3/9	2/10
Age in years, mean ± SD	74.4 ± 8.4	74.7 ± 8.7	74.1 ± 8.5
**Resting Hemodynamics**, mean ± SD			
Systolic blood pressure, mmHg	143.6 ± 11.9	143.6 ± 12.6	143.7 ± 11.8
Diastolic blood pressure, mmHg	81.4 ± 8.6	81.5 ± 9.3	81.4 ± 8.3
Heart rate, beat/min	81.5 ± 13.3	81.7 ± 10.9	81.2 ± 15.7
**Body composition**, mean ± SD			
Height, m	1.61 ± 0.06	1.60 ± 0.07	1.62 ± 0.06
Weight, kg	66.4 ± 15.9	67.8 ± 15.1	64.9 ± 17.1
BMI (kg/m^2^)	25.4 ± 6.1	26.1 ± 4.9	24.6 ± 7.2
Waist-to-hip ratio	0.97 ± 0.7	1.00 ± 0.06	0.94 ± 0.08
**Fasting blood lipids, mg/dL**, mean ± SD			
Cholesterol	153.9 ± 44.1	168.2 ± 50.7	139.6 ± 32.1
LDL	91.3 ± 32.9	103.2 ± 36.7	79.3 ± 23.9
HDL	34.4 ± 8.5	35.6 ± 8.2	33.2 ± 8.9
**Functional status**, mean ± SD			
Katz index scale	2.88 ± 1.3	2.75 ± 1.0	3.0 ± 1.5
**Medical condition**, *n*			
Type 2 Diabetes	13	5	8
Hypercholesterolemia	16	8	8
Old CVA.	7	4	3
Smoking history	6	3	3

***Abbreviations*:** SD, standard deviation; N or *n*, number; LDL, low-density lipoprotein; HDL, high-density lipoprotein; BMI, body mass index; CVA, cerebrovascular accident; F, female; M, male.

**Table 2 medicina-59-01171-t002:** Effects of aerobic exercise training on resting systolic and diastolic blood pressure.

Resting Blood Pressure	Experimental Group, *n* = 12			Control Group, *n* = 12			MD (95% CI)
	Pre Mean ± SD (95% CI)	Post Mean ± SD (95% CI)	Change (95% CI)	Pre Mean ± SD (95% CI)	Post Mean ± SD (95% CI)	Change (95% CI)	
Systolic blood pressure	143.6 ± 12.6 (136.5, 150.7)	140.8 ± 10.6 (134.8, 146.8)	−2.75 ** (−7.73, 2.22)	143.7 ± 11.7 (137.1, 150.3)	143.8 ± 11.6 (137.2, 150.3)	0.17 (−4.24, 4.58)	2.91 * (1.61, 4.21)
Diastolic blood pressure	81.5 ± 9.3 (76.2, 86.7)	80.7 ± 8.8 (75.7, 85.7)	−0.83 *** (−5.81, 4.14)	81.4 ± 8.3 (76.7, 86.1)	81.9 ± 7.8 (77.5, 86.3)	0.50 (−3.91, 4.91)	1.33 * (1.16, 1.50)

***Abbreviations*:** SD, standard deviation; CI, confidence interval; MD, mean difference. *****
*p* = 0.001; ****** *p* = 0.005; ******* *p* = 0.02.

**Table 3 medicina-59-01171-t003:** Effects of aerobic exercise training on heart rate, body fat, cholesterol profile (LDL, and HDL), and functional status.

Other Outcome Variables	Experimental Group, *n* = 12			Control Group, *n* = 12			MD (95% CI)
	Pre Mean ± SD (95% CI)	Post Mean ± SD (95% CI)	Change (95% CI)	Pre Mean ± SD (95% CI)	Post Mean ± SD (95% CI)	Change (95% CI)	
Resting heart rate	81.7 ± 10.9 (75.5, 87.9)	78.1 ± 8.9 (73.1, 83.1)	−3.67 *** (−4.23, −3.10)	81.2 ± 15.7 (72.3, 90.1)	81.4 ± 14.9 (72.9, 89.8)	0.17 (−0.62, 0.96)	3.83 ** (−4.62, −3.04)
BMI, kg/m^2^	26.1 ± 4.9 (−2.59, 54.8)	25.4 ± 4.9 (22.6, 28.1)	−0.71 **** (−1.27, −0.14)	24.6 ± 7.2 (20.5, 28.7)	24.7 ± 7.2 (20.6, 28.8)	0.1 (−0.69, 0.89)	0.81 * (−1.37, −0.24)
Cholesterol	168.2 ± 50.7 (139.5, 196.9)	133.8 ± 28.2 (117.8, 149.7)	−34.0 ** (−34.5, −33.4)	139.6 ± 32.1 (121.4, 157.7)	146.2 ± 33.2 (127.4, 164.9)	6.5 (5.71, 7.29)	40.9 * (40.3, 41.5)
LDL	103.2 ± 36.7 (82.4, 123.9)	57.2 ± 16.2 (48.0, 66.4)	−46.0 ** (−46.6, −45.4)	79.3 ± 23.9 (65.8, 92.8)	83.9 ± 2.3 (82.6, 85.2)	4.2 (3.41, 4.99)	50.3 * (49.7, 50.9)
HDL	35.6 ± 8.2 (48.9, 58.2)	36.2 ± 9.2 (30.9, 41.4)	0.58 (0.01, 1.14)	33.2 ± 8.9 (28.1, 38.2)	32.5 ± 5.8 (29.2, 35.8)	−1.0 (−1.79, −0.21)	1.55 (0.98, 2.11)
Katz index	2.7 ± 1.0 (2.13, 3.27)	2.7 ± 1.0 (2.13, 3.27)	0.0 (0.0, 0.0)	3.0 ± 1.4 (2.21, 3.79)	3.0 ± 1.4 (2.21, 3.79)	0.0 (0.0, 0.0)	0.0 (0.0, 0.0)

***Abbreviations*:** SD, standard deviation; CI, confidence interval; MD, mean difference; LDL, low-density lipoprotein; HDL, high-density lipoprotein. ***** *p* = 0.001; ****** *p* = 0.002; ******* *p = 0.005;*
******** *p* = 0.009.

## Data Availability

The data presented in this study are available on request from the corresponding author.

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
