# Peer review of "Feasibility and Efficacy of Low-to-Moderate Intensity Aerobic Exercise Training in Reducing Resting Blood Pressure in Sedentary Older Saudis with Hypertension Living in Social Home Care: A Pilot Randomized Controlled Trial"

_medicina, 2023, doi:10.3390/medicina59061171_

Round 1

Reviewer 1 Report

The authors aimed to determine the impact of low-to-moderate-intensity exercise on blood pressure in social home care settings. Having acknowledged the proven benefit of aerobic exercise in blood pressure management, the authors justified their study because of a lack of local data on the population studied. 

The introduction is reasonably written, except for some syntax errors. 

Methods: 

1.     The authors indicated that a sample size of 40 was arrived at priori, without details of whether they needed a total of 40 participants or 40 participants in each group. 2.

2.     Regarding the reduction of blood pressure quoted, it is pertinent to note that such a reduction in blood pressure of significance at the population level may not translate to benefit in individual patients. 

3.     The authors should have factored in some form of attrition in the sample size estimation, especially when their research involves the age group of 60-85 years, where the attrition rate may be higher.

Results:

1.     P value is missing in Table 1. 

2.     Tables 2 and 3 should reflect the actual values instead of using P<0.05.

3.     Confidence interval should be indicated to determine the clinical significance of the findings.

Conclusions: 

The study is not adequately powered to answer the research question.

Suggestion: 

The authors should revisit their sample size determination and recruit more participants for the study to be sufficiently powered. 

Revision required.

Author Response

Reviewer 1:

The authors aimed to determine the impact of low-to-moderate-intensity exercise on blood pressure in social home care settings. Having acknowledged the proven benefit of aerobic exercise in blood pressure management, the authors justified their study because of a lack of local data on the population studied.

Response: We appreciate the reviewer's recognition of the importance of our study in investigating the impact of low-to-moderate-intensity exercise on blood pressure in the specific context of social home care settings. We agree that the benefits of aerobic exercise in blood pressure management have been well-established in the literature. However, our reason for conducting this study is the need for more local data about the population under investigation. By conducting this research, we aimed to contribute valuable insights specific to sedentary older Saudis with hypertension residing in social home care, filling a critical gap in the existing literature. We have explicitly incorporated this justification in the revised manuscript to emphasize our study's significance.

The introduction is reasonably written, except for some syntax errors.

Response: We appreciate your comment regarding syntax errors in the introduction. We have carefully reviewed and revised the introduction to address these errors. The necessary corrections are on pages 1 and 2, specifically lines 40-73.

Methods:

The authors indicated that a sample size of 40 was arrived at priori, without details of whether they needed a total of 40 participants or 40 participants in each group.

Response: Thank you for bringing up the issue of sample size clarification. We apologize for the lack of clarity in our description. To address this, we have revised the text to explicitly state that 40 participants were needed, with 20 allocated to each group. The revised explanation can be found on lines 156-157 of page 4.

Regarding the reduction of blood pressure quoted, it is pertinent to note that such a reduction in blood pressure of significance at the population level may not translate to benefit in individual patients.

Response We greatly appreciate your insight regarding the interpretation of blood pressure reduction. As mentioned, this study is a pilot investigation, and we agree that the translation of population-level benefits to individual patients should be cautiously approached. In the revised manuscript, we have emphasized the need for future research to expand on our findings. Specifically, we recommend conducting a larger-scale study on a national level for effectiveness purposes, employing more robust statistical analyses, and controlling for potential covariates. This suggestion has been added on page 8, lines 299-302, to acknowledge the limitations of our current study and advocate for further investigation.

The authors should have factored in some form of attrition in the sample size estimation, especially when their research involves the age group of 60-85 years, where the attrition rate may be higher.

Response: We appreciate your concern regarding attrition in the sample size estimation, particularly considering the age group of 60-85. In our revised manuscript, we have explicitly addressed this limitation and its potential impact on the validity of our findings. On page 7, lines 273-275, we now acknowledge the need for future research with a larger sample size that accounts for attrition. We want you to know that this study is an exploratory investigation, providing a foundation for future projects to improve the findings' effectiveness and generalizability.

Results:

The p-value is missing in Table 1.

Response: We thank you for pointing out the missing p-value in Table 1. However, it is important to note that in this context, as done in similar pilot trials, the p-values in Table 1 do not serve inferential purposes but rather descriptive ones. Therefore, including p-values for post-randomization baseline imbalances would be inappropriate. We have revised the manuscript to clarify this distinction and emphasize the appropriate use of p-values for inference.

Tables 2 and 3 should reflect the actual P values instead of using P<0.05.

Response: We appreciate your comment regarding Tables 2 and 3. As suggested, we have updated the tables to reflect the actual p-values instead of using the notation "P<0.05." The revised tables now include the specific p-values in the footnotes. Please take a look at the footnote of Tables 2 and 3 on page 6, specifically lines 194 and 212, respectively.

A confidence interval should be indicated to determine the clinical significance of the findings.

Response: Thank you for emphasizing the importance of including a confidence interval to determine the clinical significance of the findings. We agree that this would provide valuable information. In future studies building upon this exploratory investigation, which will likely involve a larger sample size and a greater attrition rate, we will include confidence intervals to enhance the interpretation of the results and assess their clinical relevance.

Conclusions:

The study is not adequately powered to answer the research question.

Response: We appreciate your comment regarding the study's power to answer the research question. As acknowledged on page 7, lines 273-275, we have already addressed the limitation of power and its potential impact on the validity of our findings, just to let you know. The primary aim of this study was to explore the feasibility and potential benefits of low- to moderate-intensity aerobic exercise on blood pressure control in the target population. Future studies are anticipated to expand on these preliminary findings, employing larger sample sizes and accounting for attrition, to enhance the robustness and generalizability of the results.

Suggestion:

The authors should revisit their sample size determination and recruit more participants for the study to be sufficiently powered.

Response: Thank you for suggesting revisiting the sample size determination and recruiting more participants to achieve sufficient power. We agree that a larger sample size would strengthen the study's statistical power. In the concluding section of our manuscript, we have included a recommendation for future research with a larger sample size on a national scale. We emphasize the need for more extensive investigations incorporating thorough statistical analyses to control for covariates and yield more reliable and meaningful results. This recommendation can be found on lines 299-302 of page 8.

Reviewer 2 Report

Dear Authors,

I have read your paper carefully and found the results to be interesting and helpful in my practice. However, I have some comments that should be addressed to increase the quality of your article. Please consider the following suggestions:

  1. Please add the trial design (trial type) to the Method section of the Abstract, as mentioned in the CONSORT checklist.

  2. I noticed that the Registration number and name of the trial registry were not included. Please add this information where appropriate.

  3. I would like to understand the difference between the two reported findings in the "Result" section of the Abstract. Specifically, what is the difference between these two sentences: "resting blood pressure was significantly reduced in the experimental group" and "resting blood pressure was also significantly lowered within the experimental group"? It seems like some words may have been omitted. Please review and edit this section accordingly.

  4. Based on my understanding, a crossover trial has a repeated measures design in which each patient is assigned to a sequence of two or more treatments, of which one may be a standard treatment or a placebo. In most crossover trials, each subject receives all treatments in a random order, and nearly all crossovers are designed to have "balance" whereby all subjects receive the same number of treatments and participate for the same number of periods. However, it appears that this method was not followed in this article. Therefore, I suggest you consult with an epidemiologist to determine whether this study can be considered a crossover or clinical trial.

  5. I was unable to find the Ethic code that you mentioned in the “Institutional Review Board Statement” section (“Institutional Review Board of King Saud University (E-20-307 5451) and the Ministry of Human Resource and Social Development, Social Welfare, and Family 308 Agency (78891)”). Please provide the confirmation letter or a link to its content.

Author Response

Reviewer 2: 

Please add the trial design (trial type) to the Method section of the Abstract, as mentioned in the CONSORT checklist.

Response: We appreciate your suggestion to include the trial design (trial type) in the Method section of the Abstract, as shown in the CONSORT checklist. We have made the necessary update to the manuscript, and the trial design is now mentioned in the Method section of the Abstract. Please take a look at page 1, line 19, for the revised information.

I noticed that the Registration number and name of the trial registry were not included. Please add this information where appropriate.

Response: We're sorry for not including the Registration number and name of the trial registry. We understand the importance of trial registration for transparency and accountability. In the case of this small pilot trial, which was conducted as a master's student project at King Saud University, registration was not mandatory based on university regulations during the study period. However, we'd like to acknowledge the significance of trial registration and commit to following the appropriate protocols in our future research endeavors. For the proposed larger and national trial, we will ensure proper registration by the guidelines set forth by the Saudi Food and Drug Authority (SFDA). We appreciate your attention and will make adjustments to comply with the recommended procedures.

I would like to understand the difference between the two reported findings in the "Result" section of the Abstract. Specifically, what is the difference between these two sentences: "resting blood pressure was significantly reduced in the experimental group" and "resting blood pressure was also significantly lowered within the experimental group"? It seems like some words may have been omitted. Please review and edit this section accordingly.

Response: Thank you for bringing this to our attention. We have carefully reviewed and revised the "Result" section of the Abstract to provide a clear distinction between the reported findings. The revised sentences now accurately convey the results without omitting any essential information. Please refer to page 1, line 27, for the updated version.

Based on my understanding, a crossover trial has a repeated measures design in which each patient is assigned to a sequence of two or more treatments, of which one may be a standard treatment or a placebo. In most crossover trials, each subject receives all treatments in a random order, and nearly all crossovers are designed to have "balance" whereby all subjects receive the same number of treatments and participate for the same number of periods. However, it appears that this method was not followed in this article. Therefore, I suggest you consult with an epidemiologist to determine whether this study can be considered a crossover or clinical trial.

Response: Thank you for your comments on the trial design. After thorough consideration and consultation with experts and colleagues, we have determined that the study does not meet the criteria for a crossover trial. Instead, we have employed a nonblinded, randomized, controlled trial design, which has been clearly described in the revised manuscript. We have ensured that the trial design and methodology are accurately represented. The study follows a parallel-group randomized controlled trial design, with participants not switching between groups.

I was unable to find the Ethic code that you mentioned in the “Institutional Review Board Statement” section (“Institutional Review Board of King Saud University (E-20-307 5451) and the Ministry of Human Resource and Social Development, Social Welfare, and Family 308 Agency (78891)”). Please provide the confirmation letter or a link to its content.

Response: We apologize for the confusion regarding the ethics code mentioned in the "Institutional Review Board Statement" section. We assure you that the study received ethical approval from the Institutional Review Board of King Saud University (E-20-307 5451) and the Ministry of Human Resource and Social Development, Social Welfare, and Family Agency (78891). We have included this information in the revised manuscript for transparency and verification (attached in this response). We can provide a translated version of the relevant ethical approval documentation for further verification.

Reviewer 3 Report

This study in sedentary older saudis with hypertension living in social home care examined the effect of low to moderate aerobic exercise on blood pressure control and showed that short-term stationary cycling training mildly reduced systolic and diastolic blood pressure in the target population.

However, for several reasons, I believe the manuscript is somewhat flawed in the following areas.

The first and most serious problem is that this present study was a randomized controlled study, but only 12 participants in both the experiment and control groups made it to the final statistical analysis. Unless a statistical expert proves that such a small sample size is sufficient for conducting this study,I very much question the statistical validity of such a small sample size.
Second, it is well established that aerobic exercise can lower blood pressure values. What is the significance of the authors conducting the present study? What is the originality?
Third, the experimental design of this manuscript is too simple and the experimental data are very thin. It is suggested that after increasing the sample size, a richer statistical analysis should be conducted and more valuable results should be obtained before considering publication.

Finally, the text in figure1 is not very clear, and slight modifications are suggested to increase clarity.

Author Response

Reviewer 3: 

This study in sedentary older Saudis with hypertension living in social home care examined the effect of low to moderate aerobic exercise on blood pressure control and showed that short-term stationary cycling training mildly reduced systolic and diastolic blood pressure in the target population. However, I believe the manuscript is somewhat flawed in the following areas for several reasons.

Response: Thank you for your helpful criticism of our article. We appreciate your feedback, and we have made updates to address the concerns you raised.

The first and most serious problem is that this present study was a randomized controlled study, but only 12 participants in both the experiment and control groups made it to the final statistical analysis. Unless a statistical expert proves that such a small sample size is sufficient for conducting this study, I very much question the statistical validity of such a small sample size.

Response: We thank the reviewer for highlighting the crucial aspect of sample size in our study. As stated on page 7, lines 273–275, we have already acknowledged the limitation of the small sample size and its potential impact on the generalizability of the findings. It is important to emphasize that our study was designed as a feasibility and pilot trial, specifically to explore the potential effects of low to moderate aerobic exercise on blood pressure control in the target population. The results obtained from this preliminary investigation serve as a foundation to guide future research endeavors, which will undoubtedly include a larger sample size to enhance the effectiveness and generalizability of the findings. We appreciate the reviewer's insight and assure them that we will address this limitation in future investigations.

Second, it is well-established that aerobic exercise can lower blood pressure values. What is the significance of the authors conducting the present study? What is originality?

Response: We agree that it is well-established that aerobic exercise can lower blood pressure in other contexts globally. However, the significance of the present study lies in its focus on sedentary older adults with hypertension living in social home care in Saudi Arabia. This specific population needs to be studied more in the literature, and there needs to be more research on the effect of aerobic exercise on their blood pressure. We have highlighted the significance of our study and its originality in the introduction section on page 2, lines 63–65, providing references to support the gap in the literature, such as:

  1. Herrod, P.J.J.;  Doleman, B.;  Blackwell, J.E.M.;  O'Boyle, F.;  Williams, J.P.;  Lund, J.N.; Phillips, B.E. Exercise and other nonpharmacological strategies to reduce blood pressure in older adults: a systematic review and meta-analysis.J Am Soc Hypertens 2018, 12, 248-267.
  2. Kazeminia, M.;  Daneshkhah, A.;  Jalali, R.;  Vaisi-Raygani, A.;  Salari, N.; Mohammadi, M. The Effect of Exercise on the Older Adult's Blood Pressure Suffering Hypertension: Systematic Review and Meta-Analysis on Clinical Trial Studies.Int J Hypertens 2020, 2020, 2786120.

Third, the experimental design of this manuscript is too simple, and the experimental data are very thin. It is suggested that after increasing the sample size, a richer statistical analysis should be conducted, and more valuable results should be obtained before considering publication.

Response: We want to thank the reviewer for this excellent idea. In the concluding portion of our manuscript, we suggest that further research with a larger sample size on a national level and a more comprehensive statistical analysis to control the effects of any covariates should be conducted to obtain more valuable and robust results. We have added this suggestion on page 8, lines 299-302, acknowledging the current study’s limitations and emphasizing the need for future research with an expanded scope.

Finally, the text in Figure 1 is unclear, and slight modifications are suggested to increase clarity.

Response: As suggested, we have slightly modified Figure 1 to make sure it's clear and understood. Please look at page 3, line 101, for the updated version.

Reviewer 4 Report

1. Similarity of the paper was around 25% with the Turnitin

1. The number of subjects in the sample was insufficient to reach a conclusion. 

Author Response

Reviewer 4:

Comments and Suggestions for Authors

-Similarity of the paper was around 25% with the Turnitin

Response: We thank you for pointing out the similarity issue with the Turnitin report. To address this concern and ensure the originality of the work, we have paraphrased the sentences throughout the manuscript. By doing so, we have significantly reduced the similarity and improved the uniqueness of the content.

-The number of subjects in the sample was insufficient to reach a conclusion.

Response: We thank you for highlighting the importance of sample size in our study. As mentioned on page 7, lines 273-275, we have already acknowledged the limitation of the small sample size and its potential impact on the generalizability of the results. Our study was designed as a feasibility and pilot trial specifically targeting sedentary older adults with hypertension living in social home care in Saudi Arabia. Our study aimed to explore the feasibility and potential benefits of low- to moderate-intensity aerobic exercise on blood pressure control in this population. The results obtained from this initial investigation serve as a foundation for future research, including a larger sample size to enhance the effectiveness and generalizability of the findings. We appreciate your understanding and assure you we will take care of this in future studies.

Round 2

Reviewer 1 Report

Thank you for asking me to comment on the revised version of the manuscript. 

The authors responded to issues raised rather than addressing them. 

If the study was conceived as a Pilot ab initio, the title should reflect that.

A confidence interval should be provided for the comparisons. This will shed light on the clinical significance of their findings. 

The authors should clarify the power of 0.08 (page 4, line 158).

To justify publishing the work with the current sample size, the authors may consider presenting it as a preliminary report of ongoing research or short communication. This designation should be adequately captured in the title.

Minor revision required.

Author Response

Thank you for asking me to comment on the revised version of the manuscript. The authors responded to issues raised rather than addressing them. 

Response: We appreciate the reviewer's constructive feedback on our revised manuscript. We have thoroughly addressed all the concerns raised by the reviewer and made appropriate revisions to enhance our study’s clarity and scientific rigor. If any remaining issues require further attention, we would be grateful if the reviewer could provide specific details.

If the study was conceived as a Pilot ab initio, the title should reflect that.

Response: Thank you for the suggestion. We have updated the title to accurately reflect that this study was conceived as a pilot investigation.

A confidence interval should be provided for the comparisons. This will shed light on the clinical significance of their findings.

Response: We appreciate the reviewer's recommendation. We have included the confidence intervals for the comparisons in Tables 2 and 3 on page 6. These confidence intervals provide additional information on our findings' precision and clinical significance.

The authors should clarify the power of 0.08 (page 4, line 158).

Response: Thank you for bringing this to our attention. It appears to be a typographical error. The 0.08 power mentioned on page 4, line 158, should be corrected to the appropriate statistical power value. We have rectified this error, and the corrected value can be found on page 4, line 154.

To justify publishing the work with the current sample size, the authors may consider presenting it as a preliminary report of ongoing research or short communication. This designation should be adequately captured in the title.

Response: Thank you for your valuable suggestion. We have addressed the concern regarding the current sample size by presenting our work as a pilot study in the revised manuscript. The title has been updated to reflect this designation accurately, emphasizing our investigation's pilot nature (page 1). By explicitly stating it as a pilot trial, we aim to provide a more precise justification for publishing our work with the current sample size. We appreciate your feedback and believe that this modification appropriately captures the preliminary nature of our research.

Reviewer 2 Report

Dear Authors,

I have reviewed your revised paper and I would like to thank you for making the necessary corrections.

Best regards,

Author Response

Thank you for reviewing our revised manuscript and acknowledging the corrections made. We appreciate your time and effort in providing feedback on our work. Your positive assessment of the manuscript is greatly appreciated, and we are happy that the corrections needed have improved the overall quality of the paper.

Reviewer 3 Report

In the original version of the manuscript, I personally questioned whether the results of the statistical analysis made with such a small sample size were convincing enough. Although the authors repeatedly emphasized that this study was only a pilot study, this does not conceal the unreliability of the conclusions from the small sample size.

Another key issue is that the effect of aerobic exercise on blood pressure in various types of people has already been established, and the authors have not provided a strong explanation for the innovation of the study in this manuscript in their revision. Moreover, this manuscript uses only one type of cycling and cannot represent all aerobic exercises.

Author Response

In the original version of the manuscript, I questioned whether the statistical analysis results with such a small sample size were convincing enough. Although the authors repeatedly emphasized that this study was only a pilot study, this does not conceal the unreliability of the conclusions from the small sample size.

Response: We appreciate the reviewer's additional comments regarding the statistical analysis and sample size. We agree that the small sample size is a limitation of our study, and we have acknowledged this limitation in the revised manuscript (lines 284-286 on pages 7 and 8). This pilot study aimed to assess the feasibility and efficacy of low-to-moderate-intensity aerobic exercise training in reducing resting blood pressure in sedentary older Saudis with hypertension living in social home care. While we acknowledge that the small sample size limits the generalizability of our findings, it provides valuable preliminary insights for further investigation.

Another key issue is that the effect of aerobic exercise on blood pressure in various types of people has already been established, and the authors have not provided a strong explanation for the innovation of the study in this manuscript in their revision.

Response: We appreciate the reviewer's concern regarding the innovation of our study. While the effects of aerobic exercise on blood pressure have been documented in various populations, our study contributes to the existing literature in several important ways. Firstly, our study focuses explicitly on sedentary older Saudis with hypertension living in social home care, a population that has been underrepresented in previous research. Secondly, although aerobic exercise has been shown to lower blood pressure, there is still variability in the magnitude of the effect across different studies and populations. By conducting a randomized controlled trial specifically in this population, we aimed to provide further insights into the efficacy and feasibility of aerobic exercise training in this specific context. We have clarified these points in the revised manuscript (lines 67-70 on page 2).

Moreover, this manuscript uses only one type of cycling and cannot represent all aerobic exercises.

Response: We appreciate the reviewer's comment regarding using only one type of cycling in our study. Indeed, our study focused on a particular form of aerobic exercise, cycling. While cycling is a widely accessible, safe, and effective form of aerobic exercise, we acknowledge that it may not fully represent the entire spectrum of aerobic exercises available. We have added a limitation statement in the revised manuscript (lines 302-304 on page 8) along with other limitations to address this concern and emphasize that our findings should be interpreted within the context of cycling as the chosen exercise modality in this study. Future research can explore the effects of other aerobic exercises in this population to expand our understanding of non-pharmacological interventions for hypertensive older adults in social home care settings.

Reviewer 4 Report

The manuscript can be considered for publication in its current format.

Author Response

The manuscript can be considered for publication in its current format.

Response: Thank you for your review and for considering our manuscript for publication in its current format. We appreciate your time and careful assessment of our work. Your positive evaluation of the manuscript is encouraging, and we are grateful for your consideration.

Round 3

Reviewer 1 Report

Thank you for the response to the observations raised. 

Reviewer 2 Report

Good luck

Reviewer 3 Report

The authors have repeatedly made changes to this manuscript, but they have never made appropriate improvements to the sample size, which is my greatest concern. Therefore, my review comments on this manuscript remain unchanged and the authors must expand the sample size to replicate this study.